# S-Detect Software vs. EU-TIRADS Classification: A Dual-Center Validation of Diagnostic Performance in Differentiation of Thyroid Nodules

**DOI:** 10.3390/jcm9082495

**Published:** 2020-08-03

**Authors:** Ewelina Szczepanek-Parulska, Kosma Wolinski, Katarzyna Dobruch-Sobczak, Patrycja Antosik, Anna Ostalowska, Agnieszka Krauze, Bartosz Migda, Agnieszka Zylka, Malgorzata Lange-Ratajczak, Tomasz Banasiewicz, Marek Dedecjus, Zbigniew Adamczewski, Rafal Z. Slapa, Robert K. Mlosek, Andrzej Lewinski, Marek Ruchala

**Affiliations:** 1Department of Endocrinology, Metabolism and Internal Medicine, Poznan University of Medical Sciences, 60-355 Poznan, Poland; patrycja.antosik@gmail.com (P.A.); anna.ostalowska95@gmail.com (A.O.); mruchala@ump.edu.pl (M.R.); 2Radiology Department II, Maria Sklodowska-Curie National Research Institute of Oncology, 02-781 Warsaw, Poland; kdsobczak@gmail.com; 3Diagnostic Imaging Department, Medical University of Warsaw, 2nd Faculty of Medicine with the English Division and the Physiotherapy Division, 03-242 Warsaw, Poland; a.kaczor@hotmail.com (A.K.); bartoszmigda@gmail.com (B.M.); rz.slapa@gmail.com (R.Z.S.); kmlosek@poczta.internetdsl.pl (R.K.M.); 4Department of Oncological Endocrinology and Nuclear Medicine, Maria Sklodowska-Curie National Research Institute of Oncology, 02-781 Warsaw, Poland; agnieszka.zylka.edu@gmail.com (A.Z.); marek.dedecjus@gmail.com (M.D.); 5Department of General, Endocrinological, and Oncological Surgery and Gastrointestinal Oncology, Poznan University of Medical Sciences, 60-355 Poznan, Poland; chirsk2@ump.edu.pl (M.L.-R.); tbanasie@ump.edu.pl (T.B.); 6Department of Endocrinology and Metabolic Diseases, Medical University of Lodz, 90-419 Lodz, Poland; zbigniewadamczewski@gmail.com (Z.A.); andrzej.lewinski@office365.umed.pl (A.L.); 7Polish Mother’s Memorial Hospital-Research Institute, 93-338 Lodz, Poland

**Keywords:** thyroid nodules, thyroid cancer, ultrasound, computer-aided diagnosis, S-Detect, EU-TIRADS

## Abstract

Computer-aided diagnosis (CAD) and other risk stratification systems may improve ultrasound image interpretation. This prospective study aimed to compare the diagnostic performance of CAD and the European Thyroid Imaging Reporting and Data System (EU-TIRADS) classification applied by physicians with S-Detect 2 software CAD based on Korean Thyroid Imaging Reporting and Data System (K-TIRADS) and combinations of both methods (MODELs 1 to 5). In all, 133 nodules from 88 patients referred to thyroidectomy with available histopathology or with unambiguous results of cytology were included. The S-Detect system, EU-TIRADS, and mixed MODELs 1–5 for the diagnosis of thyroid cancer showed a sensitivity of 89.4%, 90.9%, 84.9%, 95.5%, 93.9%, 78.9% and 93.9%; a specificity of 80.6%, 61.2%, 88.1%, 53.7%, 73.1%, 89.6% and 80.6%; a positive predictive value of 81.9%, 69.8%, 87.5%, 67%, 77.5%, 88.1% and 82.7%; a negative predictive value of 88.5%, 87.2%, 85.5%, 92.3%, 92.5%, 81.1% and 93.1%; and an accuracy of 85%, 75.9%, 86.5%, 74.4%, 83.5%, 84.2%, and 87.2%, respectively. Comparison showed superiority of the similar MODELs 1 and 5 over other mixed models as well as EU-TIRADS and S-Detect used alone (*p*-value < 0.05). S-Detect software is characterized with high sensitivity and good specificity, whereas EU-TIRADS has high sensitivity, but rather low specificity. The best diagnostic performance in malignant thyroid nodule (TN) risk stratification was obtained for the combined model of S-Detect (“possibly malignant” nodule) and simultaneously obtaining 4 or 5 points (MODEL 1) or exactly 5 points (MODEL 5) on the EU-TIRADS scale.

## 1. Introduction

Thyroid nodules (TNs) are the most frequent endocrine disorder and occur in 10–70% of the general population, with a relatively low malignancy rate of 3–10% [1]. The most challenging issue for medical practitioners is the differentiation between benign and malignant TNs. Ultrasonography (US) of the thyroid gland is commonly used by physicians and is crucial to decide on further management, such as qualification for fine-needle aspiration biopsy (FNAB) as well as the decision on a conservative or surgical approach [2,3]. However, the analysis of nodules’ composition, echogenicity, shape, orientation, margin, calcifications, presence of halo, type of vascularization, and elasticity [4] is time consuming and subject to diverse inter-observer and intra-observer variability in the multi-ultrasound descriptor assessment of thyroid nodules, with slight to substantial agreement (κ-values ranging from 0.33 to 0.61) [5]. Moreover, no single US feature or combination of features can reliably predict the malignant character of TNs [6,7]. Furthermore, the results of US strongly depend on the operator’s interpretation and experience, which implies US sensitivity and specificity varying from 52% to 81% and 54% to 83%, respectively [8,9]. In some thyroid pathologies, additional imaging methods may be applied together with US to increase its diagnostic value [10,11]. Difficulties with the interpretation of US findings often lead to redundant FNAB or even diagnostic surgery. Recently, the European Thyroid Imaging Reporting and Data System (EU-TIRADS) was established to facilitate standardization and provide a simple lexicon for distinguishing between benign and malignant TNs and to reduce the number of unnecessary invasive interventions (FNAB, thyroidectomy) [12]. A meta-analysis of seven studies, evaluating 5672 thyroid nodules, indicated that stratifying the risk of thyroid nodules by EU-TIRADS showed high performance, while the prevalence of malignancy in EU-TIRADS class 5 was equal to 76.1% [13]. Nonetheless, even this classification might be intricate for inexperienced physicians [8]. To overcome this difficulty, computer-aided diagnosis (CAD) systems are gaining interest for US image analysis and are developed on the basis of statistical data mining algorithms, collected from medical center databases. It enables non-invasive judgement on benignity or malignancy of TNs on the basis of US image analysis [14,15]. The purpose of CAD is to increase the diagnostic confidence, achieve interpretation constancy of US features, and eliminate the inter-observer variability in order to increase diagnostic accuracy, especially when the examination is performed by ultrasonographers outside referenced centers for thyroid cancer diagnostics. The practical potential of CAD has been suggested in a few recent studies [8,9,14,16]. Kwak TIRADS has high sensitivity and low specificity. Thus, it is very useful for discarding the benign cases and reducing the number of biopsies [17]. CAD is supposed to be an applicable tool in TN diagnostics and clinical decision-making for medical practitioners with basic US skills [15,16]. It may be useful especially for less experienced operators, increasing the specificity of the examination [18]. However, there is an unmet need for validation of the method on a large cohort of patients in highly referenced centers to provide reliable information on its real clinical utility and identify a target group of physicians who may benefit from the CAD-supported evaluation of TNs. Thus, the aim of our study was to compare CAD diagnostic performance with the state-of-the-art thyroid nodule classification EU-TIRADS applied by physicians based on US morphological features, in order to support decision-making regarding the further management of TNs.

## 2. Experimental Section

### 2.1. Patients

The studied group consisted of 88 patients with thyroid nodular disease. Lesions included in the study comprised nodules whose character was verified by the unambiguous diagnostic result of a fine-needle aspiration biopsy (concerning nodules presenting category II, V, or VI according to the Bethesda classification, unless the histopathological verification was performed) or nodules in patients subjected to a thyroidectomy, for whom the result of histopathological examination was available. The **exclusion criteria** for the study were as follows: -completely cystic lesions,-lesions with eggshell calcifications,-lesions with indeterminate (category III or IV according to the Bethesda classification) or non-diagnostic cytology results (category I according to the Bethesda classification), if the histopathological verification was not performed.

The diagram depicting excluding factors and the process of recruitment of the patients for the study is presented in Figure 1.

### 2.2. Methods

This was a prospective study. Patients eligible for the study were recruited from those admitted to the tertiary reference endocrine or surgical centers for urgent evaluation of the indications for thyroidectomy due to the following results of the Bethesda System for Reporting Thyroid Cytopathology (BSRTC): (a) suspicion of malignancy or thyroid cancer (BSRTC category IV-VI or (b) nodular goiter with clinical symptoms (BSRTC category II). Nodules with category IV, V, or VI were ultimately subjected to thyroidectomy and the final result of histopathological examination served as a source of final diagnosis on the character of the lesions. In the case of patients for whom the result of the biopsy was consistent with benign lesions and who did not present with local symptoms (large goiter, compressive symptoms, dysphagia, dysphonia), they were not subjected to surgical procedure and the result of the biopsy served as the final verification of the character of the lesions.

#### 2.2.1. Ultrasound Examination

The ultrasound examination of the thyroid was performed once more on admission to the department of surgery just before the thyroidectomy or during visit in an outpatient clinic after the results of the biopsy were obtained or during the qualification for thyroidectomy. The ultrasound examinations were carried out using a Samsung RS80 EVO device (Samsung Medison, Seoul, Korea) with a 3–12 MHz linear array probe (L3-12A). CAD in our study was based on K-TIRADS classification and using S-Detect 2 software (Samsung Medison Co. Ltd., Seoul, South Korea). The following ultrasound features were assessed automatically: composition (partially cystic, solid, or cystic), echogenity (hyper/isoechoic or hypoechoic), orientation (parallel or non-parallel), margin (ill-defined, well-defined smooth, or microlobulated/spiculated), spongiform (appearance or nonappearance), shape (ovoid to round or irregular), and calcifications (microcalcification, macrocalcification, or no calcification), which were automatically assessed by the US device. Elasticity and vascularity were not assessed by CAD and have not been taken into consideration for risk stratification, because they had to be introduced manually, while for our study we intended to check the performance of CAD without being distracted by any features that were operator-dependent. Each nodule was eventually classified by S-Detect into two categories: possibly benign and possibly malignant.

The ultrasound examinations were conducted at the Department of Endocrinology, Metabolism, and Internal Medicine of Poznan University of Medical Sciences in Poznan, Poland, and at the Department of Oncological Endocrinology and Nuclear Medicine at Maria Sklodowska-Curie National Research Institute of Oncology in Warsaw, Poland. During the ultrasound examination, the transverse and longitudinal planes for both the gland and the nodules were obtained while the patient was in the supine position. The anteroposterior, transverse, and longitudinal diameters of the nodules were measured on frozen images during examination and then archived.

The examinations were performed by four physicians with at least 10 years of experience in thyroid ultrasound examination and working at the departments specialized in diagnostics and treatment of thyroid cancer. The nodules were classified with an EU-TIRADS score independently by two physicians on the basis of recorded examinations. In seven cases, there was disagreement with the score by one point. The consensus on the final grade was made together after discussion. Physicians evaluating the EU-TIRADS score were blinded to the biopsy results and/or histopathological examination. All nodules were scored according to the European Thyroid Association Guidelines for Ultrasound Malignancy Risk Stratification of Thyroid Nodules in Adults [7]. EU-TIRADS was used in a pattern-based model, where we assessed only the above-mentioned ultrasound features.

#### 2.2.2. Statistical Analysis

The calculations were performed with Statistica 12 (TIBCO Software Inc., Palo Alto, CA 94304, USA). The *p*-value < 0.05 was considered significant. Sensitivities, specificities, positive and negative predictive values, and diagnostic accuracies were calculated to assess S-Detect and the EU-TIRADS scale. Comparisons of the diagnostic values of the tests were done using McNemar’s test. EU-TIRADS classes 4 and 5 were considered as positive, whereas classes 1 to 3 were considered as negative (MODELS 1–3); in MODELS 4 and 5, only EU-TIRADS class 5 was considered as positive. Moreover, “possible malignant” by S-Detect corresponded to a positive result, whereas “possibly benign” corresponded to a negative result. The histopathological and cytological examinations served as a reference for a true or a false result.

#### 2.2.3. Ethical Approval

The study was approved by the institutional bioethical review board of each participating institution where the patients were examined, namely Poznan University of Medical Sciences and Maria Sklodowska-Curie National Research Institute of Oncology, both in Poland. All procedures were in accordance with the ethical standards of the institutional and/or national research committee and with the 1964 Helsinki Declaration and its later amendments or comparable ethical standards. Informed consent was obtained from all patients participating in the study to publish this report.

## 3. Results

Eighty-eight patients (70 women, 18 men) aged 49.4 ± 15.5 (17–80) years with 133 thyroid lesions were included in the study; of these, fifty-eight patients were diagnosed with 66 malignant lesions (including 57 papillary thyroid cancers (PTCs), four follicular variants of PTC, two medullary thyroid cancers (MTCs), one follicular thyroid cancer (FTC), two nodules in one patient described as poorly differentiated thyroid cancer). Benign lesions were colloid or hyperplastic nodules. The results of the diagnostic value of CAD-based S-Detect and physician-based EU-TIRADS (lesions with the score 4 or 5 were regarded as suspected for malignancy) are presented in Table 1. There was a statistically significant difference in the performance of CAD-based S-Detect and physician-based EU-TIRADS (*p* = 0.009).

Next, we structured five models using both CAD-based S-Detect and physician-based EU-TIRADS (Table 2).

**MODEL 1** assumes that:

- a suspected lesion obtains 4 or 5 points on the EU-TIRADS scale


**and**


- is classified by S-Detect as “possibly malignant”.

A significant difference between the diagnostic efficacy of MODEL 1 and EU-TIRADS (*p* < 0.0001) as well as MODEL 1 and S-Detect (*p* = 0.013) was detected. The diagnostic effectiveness of MODEL 1 is presented in Table 1 (row 4).

**MODEL 2** was based on the assumption that:

- a suspected lesion obtains 4 or 5 points on the EU-TIRADS scale


**or**


- is classified as “possibly malignant” by S-Detect (Table 1, row 5).

There was no statistically significant difference between MODEL 2 and the EU-TIRADS scale alone (*p* = 0.13), whereas MODEL 2 performed significantly worse than S-Detect itself (*p* < 0.001).

**MODEL 3** was based on the assumption that:

- a suspected lesion obtains 5 points on the EU-TIRADS scale


**or**


- obtains 3 or 4 points on the EU-TIRADS scale **and** simultaneously is classified as “possibly malignant” by S-Detect (Table 1, row 6).

**MODEL 4** assumes that:

- a suspected lesion obtains 5 points on the EU-TIRADS scale


**and**


- is classified by S-Detect as “possibly malignant” (Table 1, row 7).

There was no significant difference (*p* > 0.05) between the accuracy of MODEL 4 and the similar MODEL 1, however the sensitivity was slightly lower with better specificity.

**MODEL 5** was based on the assumption that:

- a suspected lesion obtains 5 points on the EU-TIRADS scale


**or**


- is classified as “possibly malignant” by S-Detect (Table 1, row 8).

There was no significant difference between the results of MODEL 5 and the similarly constructed MODEL 2 (*p* > 0.05).

The overall diagnostic performance of MODEL 3 is also not statistically significantly different from the EU-TIRADS scale alone (*p* = 0.26) but is slightly worse (*p* < 0.027) than S-Detect alone (Table 1, row 5). The obtained results suggest the superiority of MODEL 1 over the other two models as well as over the EU-TIRADS scale or S-Detect used separately. Table 2 summarizes the details of the approach used for each MODEL. Figure 2 presents an example of two lesions that were diagnosed in one patient and whose character was correctly predicted by S-Detect. The first lesion was given 4 points on the EU-TIRADS scale and was diagnosed as “possibly malignant” by S-Detect, whereas the final histopathological examination revealed the malignant character of the lesion (medullary thyroid cancer). The second lesion, although it may be attributed as suspected due to its taller-than-wide shape, was diagnosed as “possibly benign” by S-Detect, whereas it eventually turned out to be a benign hyperplastic nodule.

## 4. Discussion

Since 2009, when the first two classifications of TIRADS lexicons (Horvath and Park) were elaborated, there has been an ongoing struggle to improve the accuracy of the evaluation of TNs that would result in the adequate referral for FNAB [19]. This problem is very complicated in nature due to the diverse ultrasound features of malignant TNs and also to their different oncological aggressiveness. The performance of different classification systems also depends on the population groups being examined (i.e., oncological patients vs. primary center patients). We presume that our study satisfies the high demand for evaluating the new and promising systems and the combination of these systems for TN differentiation. To the best of our knowledge, this study constitutes the unique dual-center analysis comparing the diagnostic performance of S-Detect software and the EU-TIRADS classification in predicting the character of TNs. EU-TIRADS was established to improve inter-observer reproducibility and to facilitate communication between medical physicians [7,20]. Recent studies have demonstrated that, when using the EU-TIRADS classification, the sensitivity is satisfactory, but specificity might still be insufficient. Dobruch-Sobczak et al. have reported 98.7% sensitivity, 39.8% specificity, 38.0% PPV, and 98.8% NPV in nodules assessed as EU-TIRADS ≥4 [12]. Comparable results were obtained by Schenke et al. for small thyroid nodules (<10 mm) and presented as follows: 97.4%, 49.3%, 67.9%, and 94.4%, respectively [21]. However, Skowronska et al. indicated the following values of the mentioned parameters: 75.0%, 94.1%, 75.0%, and 94.1%, respectively, taking into account a similar threshold for the EU-TIRADS score as indicating suspected lesions [22]. In our current prospective study, the obtained value for sensitivity was high and equal to 90.9%. The specificity was moderately satisfactory and reached 61.2%. The results obtained for PPV and NPV were acceptable and similar to those in previous studies indicating that PPV of the scale is lower than NPV (69.8% and 87.2%, respectively). In this context, EU-TIRADS should be used rather to exclude malignancy (or indicate the TNs of very low probability of thyroid cancer), and those nodules might be only followed up without biopsy. On the other hand, lesions obtaining a high score on this scale should be qualified for biopsy, which is still considered as the gold standard of TN diagnostics and the most important basis of clinical decision-making in patients with TNs.

Differentiation between benign and malignant TNs is challenging, even for experienced specialists. Accordingly, some recent studies have focused on S-Detect system efficacy in comparison to evaluations performed by experienced medical physicians or other types of evaluations using CAD [14,15]. A recent meta-analysis performed by Zhao et al. including 723 lesions from five studies revealed that the sensitivity of the CAD was comparable to the assessment provided by experienced radiologists (0.87 vs. 0.88) but presented lower specificity (0.79 vs. 0.92). However, there is a need for larger sample-size prospective studies to verify the clinical usefulness of CAD in a clinical setting, together with further technical improvements of the software [15].

S-Detect for thyroid is a newly developed advanced technological tool aimed at improving the non-invasive classification of TNs. This CAD is alternatively based on the Korean Thyroid Imaging Reporting and Data System (K-TIRADS), the Russian TIRADS (RUSS) and the American Thyroid Association (ATA) guidelines. Recent meta-analyses have been focused on considerable discrepancies across ultrasound risk stratification systems and their diagnostic performance. The overall diagnostic performance of the four US-based risk stratification systems was comparable [23]. The present meta-analysis found a higher performance of American Collage of Radiology TIRADS (ACR-TIRADS) in selecting thyroid nodules for FNAB. However, the comparison across the most common US reporting systems was limited by the data available. Further studies are needed to confirm this finding [13].

Gitto et al. in their retrospective study, involving 62 patients, compared the diagnostic performance of CAD and that obtained by radiologists. The sensitivity was relatively low for CAD and equal to 21.4%, and was significantly lower than the sensitivity obtained by radiologists (78.6%). In contrast, our study demonstrated high sensitivity of S-Detect (89.4%), comparable to the assessment made by experienced sonographers using the EU-TIRADS scale (90.9%). With respect to specificity, no significant difference between CAD and radiologists was observed [24]. In our study, the specificity of S-Detect (80.6%) was also quite high and better than that of the experienced sonographers (61.2%). A high NPV of S-Detect (78.0%) together with a low PPV (25.0%) were indicated by Gitto et al. [24]. We obtained a slightly higher value for NPV; however, PPV was much more satisfactory according to our results (81.9%). In our group, assessing the malignancy risk using the EU-TIRADS scale had a lower PPV and a similar NPV in comparison to S-Detect. In terms of diagnostic accuracy, Gitto et al. observed quite similar values for S-Detect and radiologists (67.7% and 69.4%, respectively). According to our results, both values were higher (85% for S-Detect and 75.9 for endocrinologists using the EU-TIRADS scale). However, the accuracy was even better when S-Detect was combined with the EU-TIRADS assessment (86.5%). In contrast, Kim et al. in their retrospective analysis encompassing 218 nodules in 106 patients showed similar sensitivity (81.4% vs. 84.9%), but lower specificity (68.2% vs. 96.2%) of S-Detect 2, if compared to radiologists [25]. In agreement with our study, PPV of S-Detect was lower than NPV [25]. Other clinical studies showed comparable results of sensitivity (88.6%–92.0%) and specificity (74.6–87.9%) if compared to our results [26,27,28,29]. In the literature, PPV ranges from 57.5% to 89.2% [26,27,28,29], which is in agreement with our results (81.9%). We indicated slightly lower, but still satisfactory NPV (88.5%) than previous authors (90.4–98.4%) [26,27,28,29]. The discordant results, that is, low specificity (41.2%) and high sensitivity (90.5%) in the detection of malignant TNs, was reported by Xia et al. in their prospective study performed in a specialized thyroid center including 180 lesions [30]. In another study by Yoo et al. including 50 patients with 117 nodules, the authors indicated that the diagnostic performance of CAD was similar to that of radiologists [9]. It is worth noting that, similarly to our study, the authors indicated a quite high PPV (83.3%). In addition, CAD-assisted radiologists performed better in terms of sensitivity than radiologists alone [9]. This is in agreement with our study, which indicated that the combined model of taking into consideration both S-Detect and the EU-TIRADS scale assessment is better than either approach alone.

The only study performed in a similar population to ours (but including a lower number of lesions) was the one by Barczyński et al. They found that the overall accuracy of S-Detect was 82% if compared to 76% for the assessment by a surgeon with basic skills. However, the authors simultaneously indicated that CAD was inferior to an expert surgeon, due to six false-positive results [16]. A similar conclusion can be derived from the study by Chung et al. [27], where CAD performed better than a less experienced radiologist, while at least as good as an experienced physician. The summary of previous studies that evaluated the diagnostic performance of S-Detect is presented in Table 3. The results of our prospective study indicated that the evaluation of TNs based on EU-TIRADS is associated with comparable sensitivity (90.9% vs. 89.4%), but much lower specificity (61.2% vs. 80.6%) than analysis performed by S-Detect. In terms of PPV and NPV, the former is significantly better for S-Detect (81.9% vs. 69.8%), while NPV is comparable for EU-TIRADS and S-Detect used separately (87.2% vs. 88.5%). However, the combination of S-Detect and EU-TIRADS allows to maintain high sensitivity (84.9%), while, importantly, it improves specificity (up to 88.1%) as obtained for MODEL 1 in comparison to EU-TIRADS itself (61.2%). Thus, we can conclude that the evaluation of TNs using the EU-TIRADS scale by an experienced sonographer and the simultaneous evaluation by S-Detect may be complementary methods and should optimally be used together.

The lack of the histopathological verification of 35 nodules should be mentioned here as a potential limitation of the study. Of these, 17 were not subjected to surgical procedure due to a benign cytological character, an unsuspicious ultrasound picture, and a lack of local symptoms (large goiter, compressive symptoms, dysphagia, dysphonia).

## 5. Conclusions

S-Detect software characterizes with high sensitivity and good specificity, and may thus be useful in screening for suspected lesions by less experienced sonographers. The EU-TIRADS classification allows to identify suspected lesions with high sensitivity but rather low specificity. The best diagnostic performance in malignant thyroid nodule (TN) risk stratification was obtained for the combined model of S-Detect (“possibly malignant” nodule) and simultaneously obtaining 4 or 5 points (MODEL 1) or exactly 5 points (MODEL 5) on the EU-TIRADS scale.

## Figures and Tables

**Figure 1 jcm-09-02495-f001:**
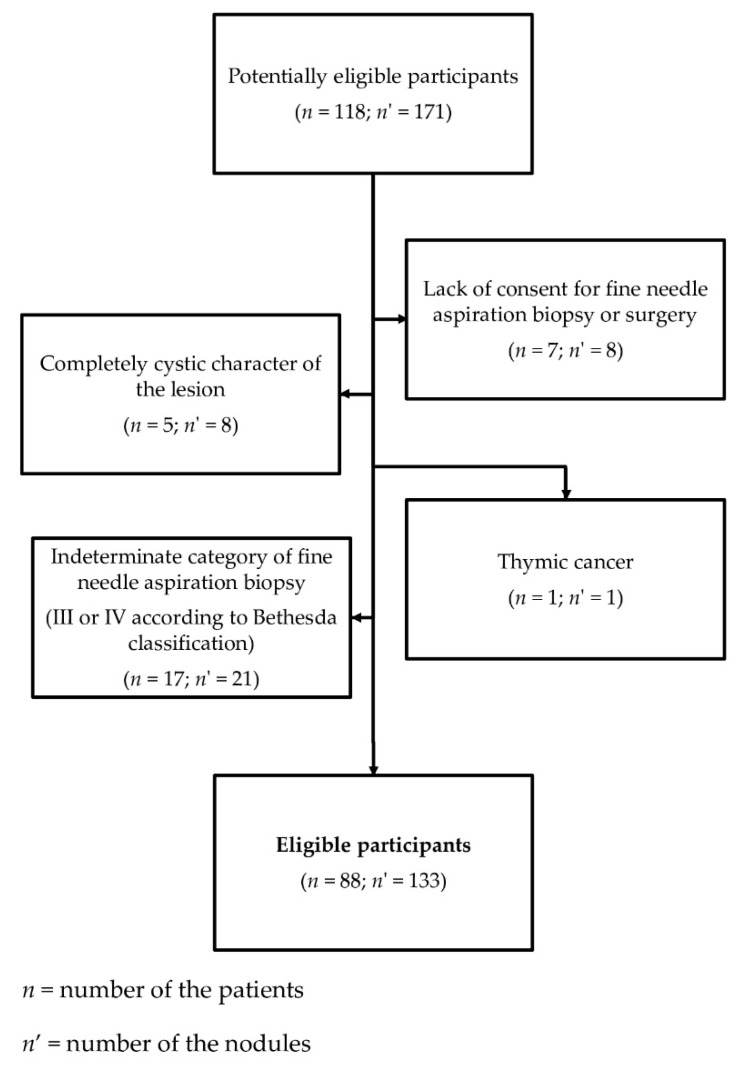
Diagram presenting the process of recruitment of the patients for the study.

**Figure 2 jcm-09-02495-f002:**
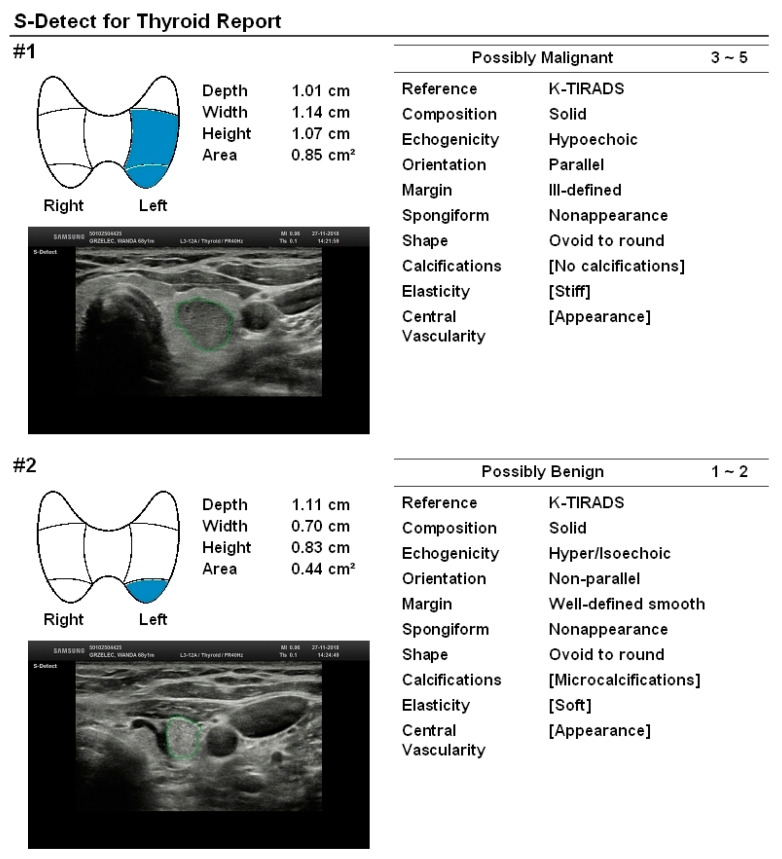
Two lesions diagnosed in one patient. The first lesion obtains 4 points on the EU-TIRADS scale, was diagnosed as “possibly malignant” by S-Detect, and the final histopathological examination revealed the malignant character of the lesion (medullary thyroid cancer). The second lesion, although it may be attributed as suspected due to its taller-than-wide shape was diagnosed as “possibly benign” by S-Detect, whereas it eventually turned out to be a benign hyperplastic nodule. Only features assessed automatically by the device, as the operator-independent ones were taken into consideration in the S-Detect classification. The two last features—elasticity and central vascularity—were not included in the S-Detect prediction. K-TIRADS, Korean Thyroid Imaging Reporting and Data System.

**Table 1 jcm-09-02495-t001:** Results for the assessment of thyroid lesions using S-Detect only, the European Thyroid Imaging Reporting and Data System (EU-TIRADS) scale (4 and 5 points), and the mixed MODELs 1 to 5.

	Sensitivity		Specificity		PPV		NPV		Accuracy	
Groups	Value	95% CI	Value	95% CI	Value	95% CI	Value	95% CI	Value	95% CI
S-Detect	89.4	79.4–95.6	80.6	69.1–89.2	81.9	73.5–88.2	88.5	79.1–94.0	85.0	77.7–90.6
EU-TIRADS (4 and 5 points)	90.9	81.3–96.6	61.2	48.5–72.9	69.8	62.9–75.9	87.2	75.7–93.8	75.9	67.8–82.9
EU-TIRADS (5 points)	80.0	68.7–88.6	79.4	67.9–88.3	80.0	71.2–86.6	79.4	70.4–86.2	79.7	72.0–86.1
MODEL 1	84.9	73–92.5	88.1	77.8–94.7	87.5	78.4–93.1	85.5	76.8–91.3	86.5	79.5–91.8
MODEL 2	95.5	87.3–99.1	53.7	41.1–66.0	67.0	61.0–72.6	92.3	79.5–97.4	74.4	66.2–81.6
MODEL 3	93.9	85.2–98.3	73.1	60.9–83.2	77.5	69.8–83.7	92.5	82.4–97.0	83.5	76.0–89.3
MODEL 4	78.9	67.0–87.9	89.6	79.7–95.7	88.1	78.5–93.8	81.1	72.8–87.3	84.2	76.9–90.0
MODEL 5	93.9	85.2–98.3	80.6	69.1–89.2	82.7	74.5–88.6	93.1	83.8–97.2	87.2	80.3–92.4

CI, confidence interval; PPV, positive predictive value; NPV, negative predictive value; EU-TIRADS, European Thyroid Imaging Reporting and Data System.

**Table 2 jcm-09-02495-t002:** Description of the mixed models. The S-Detect classification was based on computer-aided diagnosis (CAD). The nodules were classified with an European Thyroid Imaging Reporting and Data System (EU-TIRADS) score independently by two physicians on the basis of recorded examinations.

	EU-TIRADS Scale		S-Detect Classification
MODEL 1	4 or 5 points	AND	“possibly malignant”
MODEL 2	4 or 5 points	OR	“possibly malignant”
MODEL 3	5 points	-	-
	3 or 4 points	AND	“possibly malignant”
MODEL 4	5 points	AND	“possibly malignant”
MODEL 5	5 points	OR	“possibly malignant”

**Table 3 jcm-09-02495-t003:** Comparison of diagnostic performance of the computer-aided diagnosis (CAD) system and experienced staff.

	Number of Patients	Number of Nodules	Sensitivity (%)		Specificity (%)		PPV (%)		NPV (%)		Accuracy (%)	
			CAD	Staff	CAD	Staff	CAD	Staff	CAD	Staff	CAD	Staff
**Current study *, ****	**88**	**133**	**89.4**	**90.9**	**80.6**	**61.2**	**81.9**	**69.8**	**88.5**	**87.2**	**85.0**	**75.9**
Gitto S et al.	62	62	21.4	78.6	81.3	66.7	25.0	40.7	78	91.4	67.7	69.4
Kim HL et al. *	106	218	81.4	84.9	68.2	96.2	62.5	93.6	84.9	90.7	73.4	91.7
Jeong EY et al.	85	100	88.6	84.1	83.9	96.4	81.3	94.9	90.4	88.5	86.0	91.0
Chung SR et al. **	197	197	92.0	84.0	87.9	97.9	57.5	87.5	98.4	97.2	88.5	95.8
Park VY et al.	265	286	90.4–91.0	94.2	58.5–80.0	76.9	72.3–84.5	83.1	83.5–88.1	91.7	75.9–86.0	86.4
Choi YJ et al.	89	102	90.7	88.4	74.6	94.9	72.2	92.7	91.7	91.8	81.4	92.2
Xia S et al.	171	180	90.5	81.1	41.2	88.5	63.2	6.7	79.5	95.9	67.2	60.9

* CAD, only S-Detect 2 was taken into account. ** Staff with ≥7 years of experience. CAD, computer-aided diagnosis; PPV, positive predictive value; NPV, negative predictive value.

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
