# Peer review of "S-Detect Software vs. EU-TIRADS Classification: A Dual-Center Validation of Diagnostic Performance in Differentiation of Thyroid Nodules"

_jcm, 2020, doi:10.3390/jcm9082495_

Round 1

Reviewer 1 Report

The authors performed the prospective study and compared the diagnostic performance of computer-aided diagnosis (CAD) and EU-TIRADS classification of thyroid nodules. They obtained the best diagnostic performance in malignant TN risk stratification for combined model of S-detect and simultaneously gaining 4-5 points in EU-TIRADS. The paper is well written, presentation of the study is clear, methodology is appropriate, results are relevant, the conclusions are estimated on the basis of results. Please check:

# Figure 1. Please correct the spelling error "interdeminate".

# The authors said, that the nodules were classified in EU-TIRADS score independently by two physicians on the basis of recorded examinations. There was some disagreement between them, in how many of nodules the disagreement was observed?

Author Response

Poznań, 23rd July 2020

Dear Editor,

I wish to resubmit the corrected manuscript entitled“S-detect software vs. EU-TIRADS classification - a dual-centre validation of diagnostic performance in differentiation of thyroid nodules” for re-evaluation. We would be very grateful if your journal reconsidered evaluation of our efforts and potential publication of the manuscript in the revised form.

At first, authors would like to thank very much the Editor and Reviewers for the quick review as well as helpful and detailed suggestions, giving us the opportunity to improve the presentation of our work. Authors did their best to correct the paper by including all of the suggested changes in the amended version of the manuscript. Below we enclose each of the Reviewer’s comments (in italic) and our responses. Additionally, all the corrections made in the revised version of the manuscript were highlighted to expedite evaluation.

Reviewer #1: The authors performed the prospective study and compared the diagnostic performance of computer-aided diagnosis (CAD) and EU-TIRADS classification of thyroid nodules. They obtained the best diagnostic performance in malignant TN risk stratification for combined model of S-detect and simultaneously gaining 4-5 points in EU-TIRADS. The paper is well written, presentation of the study is clear, methodology is appropriate, results are relevant, the conclusions are estimated on the basis of results. Please check:

1) Figure 1. Please correct the spelling error "interdeminate".

Re: Thank you very much for this remark. Indeed there was a mistake in the figure. “Interdeminate” has been changed for “indeterminate ” and the corrected version of the scheme was placed in the text.

2) The authors said, that the nodules were classified in EU-TIRADS score independently by two physicians on the basis of recorded examinations. There was some disagreement between them, in how many of nodules the disagreement was observed?

Re: The nodules were classified in EU-TIRADS score independently by two physicians on the basis of recorded examinations. In seven cases there was disagreement in the score by one point. The final consensus was made together after discussion. The information was added to the text.

By signing this letter I acknowledge that all authors have seen and approved the revised version of the paper being submitted. I hereby confirm that no part of the submitted work has been published or is under consideration for publication elsewhere. Moreover, I confirm that there is no financial or other relationships to declare that might lead to a conflict of interest

Yours sincerely,

Kosma Wolinski & co-authors

Department of Endocrinology, Metabolism and Internal Medicine

University of Medical Sciences

49 Przybyszewskiego St., 60-355 Poznan, Poland

Reviewer 2 Report

in the presented manuscript the authors performed a prospettive study in which the diagnostic performance of computer-aided diagnosis and EU-TIRADS classification are compared. to this end 133 thyroid nodules from 88 patients were analyzed. the included patients were either referred to thyroidectomy (histopathology was avalaible), or presented unambiguous results of cytology.

the analysis was performed with 5 methods total: S-detect software alone, EU-TIRAD and three mixed models (MODEL 1, MODEL2, MODEL3).

MODEL 1 resulted to be the best one.

overall the manuscript is fine, as a suggestion, I believe it could help the reader if the authors added a table or a scheme where the mixed models are described.

Author Response

Poznań, 18th July 2020

Dear Editor,

I wish to resubmit the corrected manuscript entitled“S-detect software vs. EU-TIRADS classification - a dual-centre validation of diagnostic performance in differentiation of thyroid nodules” for re-evaluation. We would be very grateful if your journal reconsidered evaluation of our efforts and potential publication of the manuscript in the revised form.

At first, authors would like to thank very much the Editor and Reviewers for the quick review as well as helpful and detailed suggestions, giving us the opportunity to improve the presentation of our work. Authors did their best to correct the paper by including all of the suggested changes in the amended version of the manuscript. Below we enclose each of the Reviewer’s comments (in italic) and our responses. Additionally, all the corrections made in the revised version of the manuscript were highlighted to expedite evaluation.

Reviewer #2: in the presented manuscript the authors performed a prospettive study in which the diagnostic performance of computer-aided diagnosis and EU-TIRADS classification are compared. to this end 133 thyroid nodules from 88 patients were analyzed. the included patients were either referred to thyroidectomy (histopathology was avalaible), or presented unambiguous results of cytology.the analysis was performed with 5 methods total: S-detect software alone, EU-TIRAD and three mixed models (MODEL 1, MODEL2, MODEL3).MODEL 1 resulted to be the best one. Overall the manuscript is fine, as a suggestion:

1) I believe it could help the reader if the authors added a table or a scheme where the mixed models are described.

Re: We are very grateful for the suggestion. We have added a table depicting clearly the adopted approach in each of mixed models.

By signing this letter I acknowledge that all authors have seen and approved the revised version of the paper being submitted. I hereby confirm that no part of the submitted work has been published or is under consideration for publication elsewhere. Moreover, I confirm that there is no financial or other relationships to declare that might lead to a conflict of interest

Yours sincerely,

Kosma Wolinski & co-authors

Department of Endocrinology, Metabolism and Internal Medicine

University of Medical Sciences

49 Przybyszewskiego St., 60-355 Poznan, Poland

Reviewer 3 Report

Major revision

  1. Experimental section – ultrasound examination: The authors stated “CAD in our study was based on Kwak TIRADS classification and with the use of S-detect 2 software. The following ultrasound features were assessed automatically: composition (partially cystic, solid or cystic), echogenity (hyper/isoechoic or hypoechoic), orientation (parallel or non-parallel), margin (ill-defined, well-defined smooth or microlobulated / spiculated), spongiform (appearance or nonappearance), shape (ovoid to round or irregular), which were automatically assessed by the US device”. According to Kwak TIRADS five suspicious US features should be assessed: 1) composition, solid composition = 1, mixed composition = 0; 2) echogenicity, hypoechogenicity = 1, other echogenicity = 0, marked hypoechogenicity = 1; 3) margins, microlobulated margin = 1, well-circumscribed or irregular margin = 0, irregular margin = 1; 4) calcifications, microcalcification = 1, macrocalcification or no calcification = 0, macrocalcification = 1; 5) shape, taller-than-wide shape = 1, wider-than-tall shape = 0 [10.1148/radiol.11110206]. Were calcifications assessed in CAD?
  2. Methods – statistical analysis: Please report which assessment and comparisons were performed.
  3. Methods – statistical analysis: Being a diagnostic performance study, I suggest clearly specifying the definition of true/false positive/negatives (eg EU-TIRADS class 4 and 5 was considered as positive).
  4. Methods – statistical analysis: EU-TIRADS can be used in two ways: 1) pattern-based only, and nodules classified as class 2 to 5; 2) pattern- and cut-off based, and nodules be classified as indication for FNA or not. I suggest specifying that in the present study the pattern-based approach only was used, if so.
  5. Results: Reporting should be improved. It seems that two index tests were used: 1) CAD-based S-detect; 2) physician-based EU-TIRADS. In model 1, a lesion was considered as suspicious if it was classified as possibly malignant by S-detect and EU-TIRADS class 4 or 5 by the physician. Please revise the section, accordingly.
  6. Results: In Model 1 nodules classified as possibly malignant and class 4/5 were considered as positive. In Model 2 nodules classified as possibly malignant or class 4/5 were considered as positive. In Model 3 nodules classified as class 5 or class3/4+possibly malignant were considered as positive. Given that the prevalence of malignancy is significantly higher in class 5 compared to all the other classes [10.1530/EJE-20-0204], how would results change if nodules classified as EU-TIRADS class 5 (instead of class 4/5) were considered as positive? I suggest repeating all analyses, accordingly. Foremost, I wonder if CAD has a higher performance compared to EU-TIRADS class 5 only.
  7. Results: The authors stated “The first lesion was given 3 points in EU-TIRADS scale”. According to CAD and the image it seems to be a hypoechoic lesion without high risk features. If so, it should be classified as EU-TIRADS 4, as stated in the figure legend.
  8. Discussion: The authors stated “This CAD is alternatively based on K-TIRADS (Korean-Thyroid Imaging Reporting and Data System), RUSS (Russian TIRADS) and ATA (American Thyroid Association) guidelines.” While in methods they stated “CAD in our study was based on Kwak TIRADS classification and with the use of S-detect 2 software”. Please confirm that S-detect 2 has been developed on a different TIRADS compared to S-detect, if so.

Minor revision

  1. Introduction: Bibliography should be widened. I suggest the following articles to be cited. 10.1210/clinem/dgz170 and 10.1089/thy.2019.0812 concerning the available TIRADS, including EU-TIRADS. 10.1530/EJE-20-0204 concerning the performance of EU-TIRADS. 10.1007/s40477-020-00453-y concerning the performance of CAD compared to less experienced examiners. 10.1007/s00330-017-5230-0 concerning the performance of Kwak TIRADS.
  2. Experimental section – patients: Subjects with histology were included. Subjects without histology were included if: 1) category II or VI; 2) category III and IV with following surgery. How were nodules classified as category V managed?
  3. Line 88: Do the author mean “unless the histopathological verification was performed”?
  4. Figure 1: The flow of patients should be revised. 118 patients – (7+5+1+18 = 31 excluded patients) = 87 included subjects. In Figure 1 88 is reported.
  5. Figure 2: It seems that elasticity and vascularity were assessed too by CAD. Were these features taken in consideration for risk stratification? If so, how were findings classified?

Author Response

Poznań, 18th July 2020

Dear Editor,

I wish to resubmit the corrected manuscript entitled“S-detect software vs. EU-TIRADS classification - a dual-centre validation of diagnostic performance in differentiation of thyroid nodules” for re-evaluation. We would be very grateful if your journal reconsidered evaluation of our efforts and potential publication of the manuscript in the revised form.

At first, authors would like to thank very much the Editor and Reviewers for the quick review as well as helpful and detailed suggestions, giving us the opportunity to improve the presentation of our work. Authors did their best to correct the paper by including all of the suggested changes in the amended version of the manuscript. Below we enclose each of the Reviewer’s comments (in italic) and our responses. Additionally, all the corrections made in the revised version of the manuscript were highlighted to expedite evaluation.

Reviewer #3

Major revision

  1. Experimental section – ultrasound examination: The authors stated “CAD in our study was based on Kwak TIRADS classification and with the use of S-detect 2 software. The following ultrasound features were assessed automatically: composition (partially cystic, solid or cystic), echogenity (hyper/isoechoic or hypoechoic), orientation (parallel or non-parallel), margin (ill-defined, well-defined smooth or microlobulated / spiculated), spongiform (appearance or nonappearance), shape (ovoid to round or irregular), which were automatically assessed by the US device”. According to Kwak TIRADS five suspicious US features should be assessed: 1) composition, solid composition = 1, mixed composition = 0; 2) echogenicity, hypoechogenicity = 1, other echogenicity = 0, marked hypoechogenicity = 1; 3) margins, microlobulated margin = 1, well-circumscribed or irregular margin = 0, irregular margin = 1; 4) calcifications, microcalcification = 1, macrocalcification or no calcification = 0, macrocalcification = 1; 5) shape, taller-than-wide shape = 1, wider-than-tall shape = 0 [10.1148/radiol.11110206]. Were calcifications assessed in CAD?

Re: Thank you very much for this remark. We had accidentally omitted it. The presence of calcifications was taken into consideration during the assessment of the risk of malignancy by the CAD in our study. We have corrected the sentence.

  1. Methods – statistical analysis: Please report which assessment and comparisons were performed.

Re: Distrubution of results achieved using McNemar’s test. Other calculations concerning sensitivity, specificity etc. were done using commonly known formulas using Statistica software. Additional few sentences clarifying the methodology had been added.

  1. Methods – statistical analysis: Being a diagnostic performance study, I suggest clearly specifying the definition of true/false positive/negatives (eg EU-TIRADS class 4 and 5 was considered as positive).

Re: A sentence clearly stating what was treated as positive/negative and true/false result was added to the text (methods section).

  1. Methods – statistical analysis: EU-TIRADS can be used in two ways: 1) pattern-based only, and nodules classified as class 2 to 5; 2) pattern- and cut-off based, and nodules be classified as indication for FNA or not. I suggest specifying that in the present study the pattern-based approach only was used, if so.

Re: The pattern-based approach has been used and it was now explained in the text.

  1. Results: Reporting should be improved. It seems that two index tests were used: 1) CAD-based S-detect; 2) physician-based EU-TIRADS. In model 1, a lesion was considered as suspicious if it was classified as possibly malignant by S-detect and EU-TIRADS class 4 or 5 by the physician. Please revise the section, accordingly.

Re: It was now clearly stated that CAD-based S-detect and physician-based EU-TIRADS approaches were used.

  1. Results: In Model 1 nodules classified as possibly malignant and class 4/5 were considered as positive. In Model 2 nodules classified as possibly malignant or class 4/5 were considered as positive. In Model 3 nodules classified as class 5 or class3/4+possibly malignant were considered as positive. Given that the prevalence of malignancy is significantly higher in class 5 compared to all the other classes [10.1530/EJE-20-0204], how would results change if nodules classified as EU-TIRADS class 5 (instead of class 4/5) were considered as positive? I suggest repeating all analyses, accordingly. Foremost, I wonder if CAD has a higher performance compared to EU-TIRADS class 5 only.

Additional two mixed  models had been added. MODEL 4 assumes that lesion is suspected if was assessed as TIRADS 5 AND “possibly malignant” by S-detect, MODEL 5 – TIRADS 5 OR “possibly malignant”. MODEL 4 in comparison with similar MODEL 1 had a bit worse sensitivity with slightly better specificity, results achieved by models 2 and 5 were very similar, only few lesions were re-classified.

  1. Results: The authors stated “The first lesion was given 3 points in EU-TIRADS scale”. According to CAD and the image it seems to be a hypoechoic lesion without high risk features. If so, it should be classified as EU-TIRADS 4, as stated in the figure legend.

Re: We are very thankful that you noticed our mistake. There should be 4 points. We have corrected it.

  1. Discussion: The authors stated “This CAD is alternatively based on K-TIRADS (Korean-Thyroid Imaging Reporting and Data System), RUSS (Russian TIRADS) and ATA (American Thyroid Association) guidelines.” While in methods they stated “CAD in our study was based on Kwak TIRADS classification and with the use of S-detect 2 software”. Please confirm that S-detect 2 has been developed on a different TIRADS compared to S-detect, if so.

Re: CAD in our study was based on K-TIRADS classification and with the use of S-detect 2 software. The whole manuscript was once again checked and Kwak-TIRADS have been unified to K-TIRADS for clarification.

Minor revision

  1. Introduction: Bibliography should be widened. I suggest the following articles to be cited. 10.1210/clinem/dgz170 and 10.1089/thy.2019.0812 concerning the available TIRADS, including EU-TIRADS. 10.1530/EJE-20-0204 concerning the performance of EU-TIRADS. 10.1007/s40477-020-00453-y concerning the performance of CAD compared to less experienced examiners. 10.1007/s00330-017-5230-0 concerning the performance of Kwak TIRADS.

Re: We updated the introduction as discussion section with the suggested recent and interesting papers.

  1. Experimental section – patients: Subjects with histology were included. Subjects without histology were included if: 1) category II or VI; 2) category III and IV with following surgery. How were nodules classified as category V managed?

Re: Nodules classified as category V after FNAB were subjected to surgery. Only two nodules with category V have not been so far verified by histopathological examination. We considered them as malignant. We have added category V of lesions (to II and VI) to the specific sentence in our manuscript to elucidate our approach.

  1. Line 88: Do the author mean “unless the histopathological verification was performed”?

Re: Thank you for suggestion. We have added this part to the previous sentence for clarification.

  1. Figure 1: The flow of patients should be revised. 118 patients – (7+5+1+18 = 31 excluded patients) = 87 included subjects. In Figure 1 88 is reported.

Re: Thank you very much for this remark. There was a mistake in the figure. The number of patients was once again checked and n=18 in indeterminate category have been changed for 17.

  1. Figure 2: It seems that elasticity and vascularity were assessed too by CAD. Were these features taken in consideration for risk stratification? If so, how were findings classified?

Re: We have included this information to our manuscript. Elasticity and vascularity were not assessed by CAD (though in some patients were recorded by our sonographers for a separate analysis, not being a subject of this study). These features have not been taken into consideration for risk stratification, because they had to be introduced manually, while in our study we intended to check the performance of CAD not being distracted by any features that are operator-dependent (and experience-dependent) and need to be introduced by sonographers.

By signing this letter I acknowledge that all authors have seen and approved the revised version of the paper being submitted. I hereby confirm that no part of the submitted work has been published or is under consideration for publication elsewhere. Moreover, I confirm that there is no financial or other relationships to declare that might lead to a conflict of interest

Yours sincerely,

Kosma Wolinski & co-authors

Department of Endocrinology, Metabolism and Internal Medicine

University of Medical Sciences

49 Przybyszewskiego St., 60-355 Poznan, Poland

Round 2

Reviewer 3 Report

All comments have appropriately been addressed.